# Characterization of Human Subcutaneous Adipose Tissue and Validation of the Banking Procedure for Autologous Transplantation

**DOI:** 10.3390/ijms24098190

**Published:** 2023-05-03

**Authors:** Francesca Favaretto, Chiara Compagnin, Elisa Cogliati, Giulia Montagner, Francesco Dell’Antonia, Giorgio Berna, Roberto Vettor, Gabriella Milan, Diletta Trojan

**Affiliations:** 1Department of Medicine, Internal Medicine 3, Padova Hospital, University of Padova, 35128 Padova, Italy; 2Fondazione Banca dei Tessuti del Veneto (FBTV), 31100 Treviso, Italy; 3Unità Operativa Complessa di Chirurgia Plastica, ULSS2 Marca Trevigiana, 31100 Treviso, Italy

**Keywords:** adipose tissue, regenerative medicine, tissue banking, autologous transplantation, stromal vascular fraction, adipocytes

## Abstract

Adipose tissue (AT) is composed of a heterogeneous population which comprises both progenitor and differentiated cells. This heterogeneity allows a variety of roles for the AT, including regenerative functions. In fact, autologous AT is commonly used to repair soft tissue defects, and its cryopreservation could be a useful strategy to reduce the patient discomfort caused by multiple harvesting procedures. Our work aimed to characterize the cryopreserved AT and to validate its storage for up to three years for clinical applications. AT components (stromal vascular fraction-SVF and mature adipocytes) were isolated in fresh and cryopreserved samples using enzymatic digestion, and cell viability was assessed by immunofluorescence (IF) staining. Live, apoptotic and necrotic cells were quantified using cytometry by evaluating phosphatidylserine binding to fluorescent-labeled Annexin V. A multiparametric cytometry was also used to measure adipogenic (CD34+CD90+CD31−CD45−) and endothelial (CD34+CD31+CD45−) precursors and endothelial mature cells (CD34−CD31+CD45−). The maintenance of adipogenic abilities was evaluated using in vitro differentiation of SVF cultures and fluorescent lipid staining. We demonstrated that AT that is cryopreserved for up to three years maintains its differentiation potential and cellular composition. Given our results, a clinical study was started, and two patients had successful transplants without any complications using autologous cryopreserved AT.

## 1. Introduction

Adipose tissue (AT) is composed of a heterogeneous cellular population: a stromal fraction that includes endothelial cells, erythrocytes, fibroblasts, lymphocytes, monocyte/macrophages and pericytes, AT-stromal and stem cells (ASCs), progenitors, and mature adipocytes [1,2,3,4]. This heterogeneity enables the AT to play a variety of functions, which range from metabolic balance (both at the periphery and at the central level) [5] to immunomodulatory properties [6] and regenerative functions [7]. Moreover, the abundance and the accessibility of AT throughout the body has stimulated interest in its application in clinical practice as tissue for engineering and regenerative medicine [8]. Clinically, the transplantation of whole AT has several applications in dermatology, aesthetic surgery [9] and plastic surgery, such as soft tissue for defect correction in burns [10], ulcers [11], diabetic foot ulcers [12], postmastectomy breast reconstruction [13] and scleroderma [14]. The first studies intended for autologous fat transplantation started at the beginning of the first half of the twentieth century [15]. Later on, in the 1990s, the advancements obtained in liposuction and lipofilling techniques made it possible to apply AT transplantation routinely [16]. Until now, autologous tissue has been harvested and immediately transplanted in a one-step procedure developed mainly by Coleman et al. Even with the great improvement’s brought about by Coleman’s technique for ensuring high tissue viability [17], there are still some complications. These involve the variability in the results of lipofilling, given that graft absorption is not predictable and can reach up to 70% of the volume of the original fat graft [18]. This could represent a possible limitation for patients, especially if they are affected by chronic diseases requiring multiple injections to control the defect. Thus, through a reduction in harvesting procedures, AT cryopreservation could ameliorate a patient’s discomfort and reduce the costs for the healthcare system by maintaining the main features of the ex vivo sample. To date, authors have reported different methods to achieve fat cryopreservation that mainly differ on the cryoprotective agent (CPA) applied to prevent ice formation in the tissue. The most successful CPAs were trehalose, glycerol and dimethyl sulfoxide (DMSO) or their combinations. The efficacy was tested by in vitro/ex vivo studies and preclinical models [19], but data on reliability of AT cryopreservation are still controversial [20,21,22]. The presented protocols are also far from being applicable in humans. Trehalose and glycerol have the advantage of being less toxic when compared with DMSO; nevertheless, the latter is widely used as CPA in tissue and cell banks and in tissues composed of multiple cell types [23]. Moreover, it was reported that DMSO ensures a faster growth of mesenchymal stem cells derived from AT than trehalose [24]. Shaik et al. developed a freezing protocol at −20 °C and −80 °C replacing or reducing DMSO with polyvinylpyrrolidone (PVP). The new CPA media displayed robust adipogenic and osteogenic differentiation potential; however, further studies should be conducted to optimize the new cryopreservation protocol [25]. Moreover, the optimal temperature of AT preservation has been investigated by Kim et al., in this way demonstrating that AT should be stored at −80 °C; however, the authors did not analyze the influence of the CPA media on the reported results [26]. There are very few papers reporting a clinical application of cryopreserved AT in patients, which is used primarily for aesthetic applications [20,27,28,29]. Authors described the lack of severe complications but provided less information regarding the protocol for cryostorage and the characterization of the implanted tissue. In plastic surgery, AT grafts for breast reconstruction are conducted to obtain volume augmentation through both mature adipocytes and ASCs, with high patient and surgeon satisfaction rates and no evidence of cancer recurrence [30]. In this paper, we described our procedure to store and evaluate the cellular viability, composition and adipogenic potential of long-term cryopreserved AT intended for use in plastic surgery to correct defects in reconstructed breasts after mastectomy. We studied the storage of AT in aliquots that would avoid multiple thawing of AT in a clinical setting. Moreover, we analyzed key tissue banking topics linked to the management of cryopreserved AT, such as quality control for viability and safe delivery to clinicians. It is noteworthy that Fondazione Banca dei Tessuti del Veneto (FBTV) has obtained the approval from the competent authority to retrieve, process, cryopreserve, store and distribute AT for autologous application. Subsequently, cryopreserved AT has been transplanted in four patients to improve postmastectomy breast reconstruction.

## 2. Results

### 2.1. AT Long-Term Cryopreservation

#### 2.1.1. Cryopreserved AT Maintains Viable Stromal Vascular Fraction

Thawed samples derived from different cryopreservation timings underwent collagenase digestion following Rodbell’s modification protocol to isolate stromal vascular fractions (SVF) and mature adipocytes (AD).

As showed in Figure 1 and Appendix A, manual counting using a hemocytometer and fluorescent staining displays live cells in the SVF. As highlighted in Figure 1a, the samples derived from four different donors displayed a certain degree of variability in terms of cell content/g of tissue in each tested condition. No difference was observed among samples in terms of percentage of cell viability when comparing fresh and cryopreserved lipoaspirates (Figure 1b). Besides a tendency for a reduction in total cell number/g between fresh samples and those coming from the first month of storage (but not statistically significant), cryopreserved samples displayed a very similar trend in live cells, ranging from 65.7% to 86.7% (median value); this was independent from the time of storage (from 1 month up to 3 years). The cryopreservation method allows the maintenance of an optimal range in viability that is similar to fresh samples (Figure 1b).

#### 2.1.2. Cryopreservation of AT Does Not Affect Annexin V Nor Propidium Iodide Staining in SVF

Cell viability, apoptosis and necrosis were assessed evaluating the phosphatidylserine translocation onto the plasma membrane using fluorescent labeled Annexin V and cell permeability to propidium iodide using flow cytometry. Data were collected from four different donors in each tested condition. As represented in Figure 2a,b, cellular distribution in live, early and late apoptosis or necrosis is similar in fresh and cryopreserved samples along the storage. Live cells range from 34 to 59% (median value: 41%) in fresh samples; after storage, the lowest value we observed was 24%. The percentage of viability is different from the one reported in Figure 1, but it depends on the method used to assess it. Early apoptosis is similar between fresh lipoaspirates (median value: 35.6%) and cryopreserved tissues (median value ranging from 39 to 45%). At t2, a reduction in viability and an increase in late apoptotic cells were observed but without any significance. Overall, only a small percentage of cells were necrotic (less than 3.5%). Cell viability estimated by microscopy and cytometry positively correlated (r: 0.593, *p* < 0.01), as reported in Figure 2c.

#### 2.1.3. Cryopreserved AT Maintains the Expression of Specific Membrane Markers on SVF Cells

To better characterize the effects of cryopreservation on cell quality and immunophenotype, we used a cytofluorimetric multiparametric approach that analyzes membrane marker expression. The hemopoietic population was excluded from the analyses on the basis of CD45 positive staining. Then, we quantified the percentage of adipose-derived stromal and stem cells: ASCs as CD34+CD31−CD45−, the endothelial precursors as CD34+CD31+CD45− and the endothelial mature cells as CD34−CD31+CD45−. The presence of mesenchymal surface markers was measured by quantifying CD90 positivity. As showed in Figure 3a,b, cells derived both from fresh and preserved tissues showed similar distribution of the different subpopulations. CD34 is expressed by 33.5 to 47.5% of SVF cells, without a clear effect on storage. The percentage of precursor and mature endothelial cells is also similar among the samples. In addition, the mesenchymal marker percentage is still unaffected by the storage of the tissue in liquid nitrogen vapors (>91% in all samples analyzed, median value).

#### 2.1.4. Cryopreserved Stromal Vascular Fraction Retains In Vitro Adipogenic Abilities

The adipogenic potential of the progenitor cells included in SVF was assessed by in vitro induction of adipogenesis and quantification through immunofluorescence (Figure 4a) using an adipogenic index (number of adipocyte/total nuclei) in samples derived from four different donors. As shown in the plot (Figure 4b), adipogenic abilities were also maintained in cells derived from the cryopreserved tissue, even if with a different range of extent. We observed a reduction (*p* < 0.5 vs. t0) in adipogenic differentiation only in the samples stored for 14 months. However, samples cryopreserved for 36 months displayed similar adipogenesis when compared to fresh and short-term conservation.

#### 2.1.5. Cryopreserved AT Retains Viable AD

AD viability was estimated by manual counting using a hemocytometer and fluorescent stains highlighting live (blue nuclei) and dead (purple nuclei) (Figure 5a). Green fluorescence highlights the lipid droplets. As shown in Figure 5b, the number of adipocytes seem to be variable among samples derived from four different donors, showing a significant reduction at 14 months of storage in comparison to fresh tissue. Nevertheless, data obtained through three-year cryopreservation showed a comparable number of cells when compared to fresh and short-term conservation. These data are also confirmed by analyzing the percentage of viable cells, as shown in Figure 6c.

#### 2.1.6. Microbiological Tests

Draining fluids of AT specimens were collected before and after decontamination in antibiotic solution. The samples were inoculated in a blood culture bottle as described in the Materials and Methods section. All the microbiological tests performed on the tissues were compliant with the acceptance criteria for transplantation. The analysis performed before antibiotic decontamination of the adipose specimens did not reveal microorganisms that result in discarding the tissue according to European guidelines on the quality and safety of tissues for transplantation [31]. After antibiotic decontamination, all the samples analyzed were negative. Moreover, according to the national directives on the processing of human tissues for transplantation, environmental controls were compliant. 

### 2.2. Validation of Quality Controls and Logistic Procedures 

#### 2.2.1. Quality Control Samples Display Similar Viability and Composition of Specimens for Clinical Use

To ensure the transplantation of only high-quality tissues in patients, during the cryopreservation we set up a quality control step consisting of the preservation of a paired sample (satellite) of small volume for each adipose aliquot where cell viability and composition were verified before tissue distribution for implantation. We then analyzed the percentage of live cells measured by immunofluorescence in the samples and their satellite, as described in Figure 1. Comparable cell viability between samples in FVS (Figure 6a) and AD (Figure 6c), which positively correlated (FVS: r = 0.829, *p* < 0.001, Figure 6b; AD: r = 0.620, *p* < 0.05, Figure 6d), was observed. Moreover, the surface marker pattern was conserved between the two groups (Appendix A).

#### 2.2.2. Delivery with Dry Ice of AT at −80 °C Preserves Graft Viability 

To verify safe delivery of the tissues to the surgeons, we analyzed the effects of the shipment of frozen samples with dry ice at −80 °C on samples derived from three different donors. We then compared cell viability and composition between samples stored in vapor phase liquid nitrogen and readily thawed versus samples which were subjected to a shipment with dry ice. As noticeable in Figure 7, the two groups did not display any differences in terms of cell number or cell viability, both in SVF (Figure 7a,b) and AD (Figure 7c,d). The immunophenotype of SVF is comparable between the two groups (Appendix A).

#### 2.2.3. Temporary Storage of AT at −80 °C Preserves Graft Viability

We also studied the possibility of short-term storage of the AT at −80 °C after cryopreservation to allow the clinician to briefly keep the tissue before implantation. After preservation in vapor phase liquid nitrogen, samples from three different donors were conserved at −80 °C for 1, 2 and 3 months. As shown in Figure 8, short-lived storage at −80 °C did not affect cell number and viability and did not exhibit any difference regarding the length of the storage at −80 °C in SVF (Figure 8a,b) and AD (Figure 8c,d). However, higher variability in the viability of adipocytes was observed, especially at 3 months of storage at −80 °C. The short-time preservation at −80 °C did not impact the immunophenotype of the SVF (Appendix A).

## 3. Discussion

In surgery, autologous AT is commonly used to repair soft tissue defects [32,33]. Cryopreservation represents a useful strategy to reduce the discomfort caused by multiple procedures of harvesting and grafting of fresh tissue on a patient. There are several published data investigating the possibility to store AT, but an available and standardized procedure to preserve AT for clinical applications is still lacking. Previous studies analyzed the effects of AT cryopreservation on cell viability, investigating these properties on fixed and not on ex vivo samples. The authors provided information primarily regarding tissue structure [27], which is well maintained [24,27,29], and evaluated mature adipocytes [34] or SVF [35] one by one. Extensive studies were performed on cryopreservation of purified adipose-derived stromal and stem cells ASCs/MSC (AT) [36,37,38]. Our work aimed to characterize the cryopreservation of AT of several samples stored for up to three years in liquid nitrogen vapors, evaluating key points in the perspective of tissue banking such as microbiological contamination. Moreover, using different methods, we analyzed cell viability and the maintenance of cellular composition for both SVF and AD, evaluating the expression of specific cell surface markers and the adipogenic potential. To our knowledge, no other authors have investigated both fractions simultaneously in samples stored for up to three years [27,39] nor validated the quality and safety of the tissue for human application and tissue banking. 

The procurement, transport and processing of tissues have the potential risk of microbiological contamination; hence, microbiological testing is important for recipient safety [31,40]. The microbiological analyses we performed confirmed that our samples are devoid of contamination, indicating that the procurement step followed by a decontamination procedure gives rise to safe tissue compatible with clinical applications. To date, several groups have reported different strategies to prevent the formation of ice crystals in cells using CPAs other than DMSO. Trehalose-based CPAs were mostly used by Cui XD et al. [41] and seemed to be promising, but it did not give rise to successful results in terms of cell viability (>93% reduction t1 vs. t0, data not showed) in our hands. Glycerol (70%) has been recently reported as the best cryoprotector by Zhang P-Q et al. [42] who obtained results similar to ours in terms of cell viability after 1 month of cryopreservation (72.67 ± 5.80% vs. 65.77 ± 11.1, respectively). Unfortunately, the authors did not report the viability of the sample at t0; thus, it is not possible to evaluate the effects of the storage procedure.

The temperature used for the storage of the AT is a critical variable, and some authors analyze its effect on cell viability. Erdim M et al. compared the short-term storage of AT at different temperatures (+4 °C; −20 °C and − 80°C) and found a reduction in live adipocytes when compared to freshly isolated cells [43]. Similar results were also obtained by Wolter TP et al. who observed that the use of a CPA can improve cell survival [44]; otherwise, adipocytes would rapidly lose their viability [45]. Moreover, a controlled freezing rate and storage in liquid nitrogen vapors ensure better cryopreservation [46]. Our data showed variability between the samples we tested, which is noticeable in fresh tissues (t0) and also reported by other authors [27,47]. Even if the cryopreservation seems to partially reduce the number of viable cells per gram of tissue on the SVF, we did not observe any statistical reduction between samples, thus confirming the maintenance of live cells during the storage. This was verified using both microscopy and cytometry approaches and by the correlation among them (Figure 2c). Moreover, to overcome possible cell counting bias due to different methods and difficulties in the counting of cell clumps, we compared a well-known method for cell counting, i.e., trypan blue staining [48], with a cell counting assay using fluorescent intercalating agents to rapidly detect live and dead cells (Appendix A–d) [49]. The positive correlation we found among them (r = 0.857, *p* < 0.001, Appendix A) highlights the possibility of using this method, one that could also be automated. 

Stored AT samples also contain live adipocytes. This seems to be influenced more by the conservation than the SVF given the high variability we found in their number. Nevertheless, these data confirm the fragility of AD and agree with other studies which demonstrated that AD viability can be reduced after cryopreservation [43].

In the SVF, the detailed characterization of live and apoptotic cells obtained using Annexin V allowed us to observe that cryopreservation did not induce more cell death than in fresh lipoaspirates nor was there a different pattern distribution of cells on early and late apoptosis and necrosis (Figure 2). In humans, the combination of a few markers allows the identification and quantification of the ASC (adipose-derived stromal and stem cells) population ex vivo (CD34+/CD31−/CD45−) to study its role under different conditions [1,50,51]. The immunophenotyping of the SVF revealed a well-conserved percentage of adipocyte and endothelial precursors and of mature endothelial cells between samples. Moreover, the adipocyte precursors were still maintaining the mesenchymal marker CD90. These findings showed that cryopreservation did not alter tissue composition, suggesting the maintenance of all the features linked with the different cell types which allow the formation of adipose and vessel structures [32,52]. Furthermore, the preservation of the adipose-derived stromal and stem cells ASCs/MSC (AT) [53,54] suggests the conservation of all the abilities identified for these cells, such as immunomodulatory properties [47,55], which can help tissue engraftment, thus overcoming all the challenges related to preparation associated with the therapeutic applications of MSC [56] as advanced therapy medicinal products (ATMPs). Moreover, SVF was still conserving its adipogenic abilities, giving rise to adipocytes in vitro; this agrees with other published data [39] that also confirmed the preservation of other mesenchymal markers and lineage commitment [24,39]. The heterogeneity of adipose tissue is driven by differences in sex, genetics and environment, but also by the various depots existing in humans and their cellular composition [57]. It is well known that diversity in morphology, metabolism and function shapes metabolic status, negatively influencing the entire energy balance and controlling the appearance of metabolic diseases [58]. On the other hand, the dynamic interactions of the SVF cells with the milieu through paracrine and endocrine signaling can positively impact the local and systemic responses stimulating anti-inflammatory and proangiogenic activity [59].

Currently, it is not possible to understand how this heterogeneity can impact fat grafting; however, the autologous transfer would maintain the tissue’s patient characteristics. Moreover, differences in gas exchanges and nutrient supply can impact fat grafting in relation to the different volumes of the stored AT samples. In our work, we reported that cell viability, differentiation and subpopulation distribution are statistically unaltered by storage in smaller (satellite, 10 mL) and larger (50 mL) samples, thus supporting that, in this case, the volume of cryopreserved tissue does not have an impact.

In our study, we paid special attention to several aims related to tissue banking: In fact, we validated the possibility of carrying out a quality control (satellite sample) test for the cryopreserved AT that can help evaluate the percentage of live cells before their distribution, thus enabling the implantation of only viable tissue in patients. Moreover, we also investigated the impact of tissue transportation in dry ice and of short-term storage in an ultra-freezer (−80 °C). This ensured that delivery and preservation at higher temperatures would not influence the tissue’s properties. We clearly showed that satellite samples can be regarded as quality control samples. Moreover, shipment and short-term storage (3 months) at −80 °C did not affect AT viability nor composition, guaranteeing instead better quality and safety of the tissue for human application. In particular, the possibility brief conservation at −80 °C gives rise to the opportunity of overcoming the problem of rescheduling surgery for patients, if necessary, without discarding the tissue.

Our work has some limitations which are mainly due to the limited possibilities for collecting enough tissue and individuals to perform all the tests considered in the experimental plan. A certain degree of variability was observed between the batches that were collected, but this is similarly reported in other publications and could possibly be associated to the different anatomical source of the lipoaspirates. We could not perform a site-specific analysis given that in the clinical setting it is not applicable to procure enough starting material for preservation from a single liposuction site. We also decided to analyze the effects of cryostorage on different cellular components of AT, since there are abundant reports describing that the overall tissue structure is not affected by freezing [24,27].

It is well established that ASCs (adipose-derived stromal and stem cells) possess mesenchymal markers and multi-lineage differentiation potential [24,42]; thus, we focused our analyses on in vitro adipogenic potential, which is very useful for breast reconstruction purposes. Moreover, gene expression or functional assays of specific subpopulations were not carried out because they would have required a sorting strategy with sophisticated instruments not commonly available at hospital facilities. We did not perform in vivo preclinical studies because we obtained the approval from the Italian National Authorities for cryopreserved AT autologous application in patients.

## 4. Materials and Methods

### 4.1. Study Design

In our study, we accomplished a characterization of AT aimed at validating the procedures to implement at the FBTV.

Long-term storage without affecting tissue viability and tissue safety: AT samples were collected (t0) and stored in vapor phase liquid nitrogen for 1 (t1), 2 (t2), 3 (t3), 14 (t4) and 36 (t5) months; at each time point, the samples were analyzed and compared to the fresh tissue (t0). Moreover, microbiological tests were carried out;Quality controls before implantation: AT samples were collected and split in aliquots of 50 and 10 mL, mimicking the aliquots to transplant and paired quality control samples, respectively. The viability and composition of both groups were compared to verify this approach. The storage of AT in aliquots has been proposed to avoid recurrent thawing when multiple filling procedures are necessary in a clinical setting;Distribution procedure in dry ice: AT samples were collected and cryopreserved in aliquots. Afterwards, aliquots kept in dry ice for 24 h were compared to aliquots stored in vapor phase liquid nitrogen;Temporary storage at −80 °C: After collection, AT samples were cryopreserved in aliquots. Subsequently, some of them were kept at −80 °C for 1, 2 and 3 months and compared to samples stored in vapor phase liquid nitrogen for at least 1 month.

A scheme for the protocol is depicted in Figure 9.

### 4.2. Human Sample Fat Harvesting

Subcutaneous AT was collected from seven female donors who had undergone liposuction after bariatric surgery at the Plastic Surgery Unit of Treviso Hospital. Liposuction was conducted from the abdomen (periumbilical area), medial thighs or knees using a 4 mm suction cannula at constant minimal negative pressure under general anesthesia using a modified Klein’s solution (1:1000 epinephrin in saline solution). The tissue was then transferred to a closed-membrane filtration system for processing harvested fat (Puregraft 250 filtration system, PureGraft, Solana Beach, CA, USA) and readily sent at +4 °C to the FBTV.

### 4.3. Decontamination, Cryopreservation, Storage and Thawing Procedures

Within 8 h from liposuction, AT was decontaminated for at least 12 h at +4 °C by adding an antibiotic solution containing Vancomycin (100 μg/mL), Meropenem (200 μg/mL) and Gentamicin (200 μg/mL) in BASE medium (Alchimia, Padova, Italy) in a 1:1 ratio. The antibiotic solution was previously validated for tissue decontamination both in vitro and ex vivo [60,61,62]. Afterwards, the antibiotic solution was discarded by decanting, and the AT was washed using a saline solution in a 1:1 ratio. The AT was then aliquoted (50 mL) in sterile bags, adding the cryopreserving solution made with 10% DMSO (WAK-Chemie Medical GmbH, Steinbach/Ts Germany) and 2% human albumin (Behring, Milan, Italy) in BASE medium in a 1:1 ratio. All the procedures were carried out aseptically under an a class A laminar-flow hood in a g class B environment.

Cryopreserved samples were frozen using a programmable cryogenic freezer Planer KryoSave Integra 750–30 (Planer Limited, Sunbury-On-Thames, UK) for the controlled cooling rate.

Before analyzing, AT was thawed using a water bath for 10 min at 37 °C, rinsed two times with saline solution and then used for further analyses.

### 4.4. Characterization of AT

#### 4.4.1. Cell Isolation (SVF-AD)

AT components were collected from lipoaspirates using a modified Rodbell’s protocol as described in the Appendix II of Sanna et al. [63]: AT was directly digested in a collagenase type 2 solution (1 mg/mL, Merck, Darmstadt, Germany) in DMEM F12 (ThermoFisher Scientific, Waltham, Massachusetts, USA) for 30 min at 37 °C through gentle stirring. Then, tissue was mechanically disrupted using a pipette and the digested tissue was filtered through a nylon mesh of 100 µm pore size (Corning, NY, USA). After low speed (100× *g*) centrifugation, the upper phase was collected and stored as adipocytes (AD). The residual volume was then centrifuged at 300× *g*, and the pellet was collected as a stromal vascular fraction (SVF). SVF was then subjected to a red blood cell lysis using a hypertonic buffer. After the last centrifugation, the recovered cells were then counted.

#### 4.4.2. Cell Count and Cell Viability Quantification

The cell suspension (20 µL) was stained using a trypan Blue Solution 0.4% (Thermo Fisher Scientific, Waltham, MA, USA) performing a 1:1 dilution. Cells were then loaded in a hemocytometer and counted under light microscope (DMILED, Leica Microsystems, Mannheim, Germany).

Live and dead cells were counted using the ReadyProbes Cell Viability Imaging Kit (Blue/Red) (Thermo Fisher Scientific). The staining was performed following the manufacturer’s instructions: 500 µL of the cell suspension was incubated for 10 min with a mixture of each dye and loaded in a hemocytometer. In addition, the AD fraction was also stained with Bodipy 493/503 (Thermo Fisher Scientific) to highlight lipid droplets. Pictures were taken using a fluorescent microscope equipped with a camera (DMI6000, Leica Microsystems). Cells were manually counted, and viability was expressed as the percentage of live cells (blue) over the total. When needed, the weight of each decanted lipoaspirate was used to calculate the yield of recovered live cells per gram of tissue.

#### 4.4.3. Annexin V Staining and Analysis

Annexin-V-FLUOS Staining Kit (Merck) was used to identify apoptosis and necrosis following the manufacturer’s instructions. A quantity of 1 × 10^5^ SVF cells was centrifuged and resuspended in 400 µL of incubation buffer (HEPES Buffer) with the addition of 2 µL of annexin V and propidium iodide. The negative control was obtained using the incubation buffer only. Cells were kept 10 min in the dark at room temperature, centrifuged and resuspended in 100 µL of HEPES Buffer. The samples were acquired (20,000 events) using a CytoFlex Flow Cytometer (Beckman Coulter, Brea, CA, USA). Data acquisition and analyses were performed using Kaluza software.

#### 4.4.4. Immunophenotyping and Flow Cytometry

A quantity of 1 × 10^5^ SVF cells was washed with a cold FACS buffer (2% BSA/PBS1×) and simultaneously incubated in the dark for 10 min at room temperature with monoclonal mouse anti-human fluorochrome-conjugated antibodies (BD Biosciences) in different combinations, as described in Belligoli et al. [64] and shown in Table 1. After washing with the FACS buffer, the labeled cells were resuspended in 200 µL of FACS buffer and acquired (20,000 events/sample) using a CytoFlex Flow Cytometer (Beckman). Data acquisition and analyses were performed using Kaluza software. Regions and gates were set by negative-control, isotype-matched PE-IgG1, FITC-IgG1 and PerCP-Cy5.5-IgG1 monoclonal antibodies (BD Biosciences). ASCs (adipose-derived stromal and stem cells), identified as CD45−/31−/34+, were quantified as percentages in the SVF morphological gate. The endothelial progenitors and mature cells were identified, respectively, as CD45−/31+/34+ and CD45−/31+/34− and quantified using the same procedure. The percentage of mesenchymal marker CD90 was quantified in the immunological gate of CD34 [50,65].

SVF cells isolated from fresh and cryopreserved samples were incubated with a combination of monoclonal mouse anti-human fluorochrome-conjugated antibodies recognizing different antigens to perform multiparametric flow cytometry analysis. Isotype-matched FITC-, PE- and PerCP.Cy5.5-IgG1 monoclonal antibodies were used as negative controls. An unstained sample was used for auto-fluorescence control.

#### 4.4.5. Adipogenic Differentiation

A quantity of 0.1 × 10^6^ cells/well (96 wells plate) from the SVF were seeded in human standard medium (h-SdM): DMEM F12 supplemented with 10% FBS, 150 U/mL streptomycin, 200 U/mL penicillin, 2 mM glutamine and 1 mM HEPES (all from ThermoFisher Scientific). At cell confluence (1/2 days later), the medium was replaced with human adipogenic medium (h-AdM): DMEM F12 (with 150 U/mL streptomycin, 200 U/mL penicillin, 2 mM glutamine, 1 mM HEPES) containing 66 nM insulin, 100 nM dexamethasone, 1 nM 3,3′,5-triiodio-L-thyronine (T3), 10 µg/mL transferrin, 33 µM biotin, 17 µM pantothenate, 0.25 mM IBMX and 10 μM rosiglitazone (all from Merck). IBMX and rosiglitazone were removed after 3 days of culture. At day 20, cells were stained (fresh) using 10 ug/mL of Hoechst 33342 and Bodipy 493/503 (both from ThermoFisher Scientific). Pictures were taken using fluorescent microscope and the adipogenic index was calculated by defining the adipocyte number/total nuclei × 100.

### 4.5. Microbiological Testing

In addition to the standard environmental controls, two 10 mL samples for each draining fluid of the adipose biospecimens were inoculate into BACTEC Plus Aerobic/F, Anaerobic/F blood culture bottles (BD Biosciences, East Rutherford, NJ, USA) before the decontamination procedure and the cryopreservation of the samples.

### 4.6. Statistical Analyses

Statistical analyses were performed using SigmaPlot 14 (Systat Software Inc., San Jose, CA, USA). For multiple comparisons, ANOVA on ranks with Dunn’s or Tukey post-hoc tests were performed for the two-sample comparison t-test or the Mann–Whitney test. Data were expressed as mean ± standard deviation (SD) or median and interquartile ranges. The correlation between variables was calculated using Spearman’s correlation method.

## 5. Conclusions

We demonstrated that AT cryopreserved up to three years maintains its differentiation potential and cellular composition, keeping viable precursors and mature cells. The procedure seems to be promising for autologous AT transplantation.

## Figures and Tables

**Figure 1 ijms-24-08190-f001:**
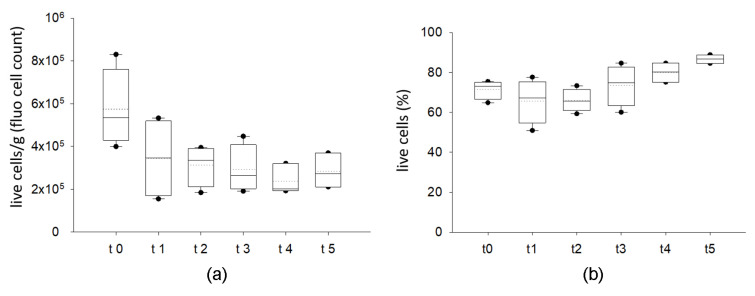
Evaluation of cell viability in the SVF (stromal vascular fraction) of fresh (t0) and cryopreserved AT (t1–t5) using microscopy (n = 4). In (**a**), number of viable cells normalized by sample weight (g) calculated using nuclear fluorescent staining for each time point. (**b**) Percentage of viable SVF calculated using nuclear fluorescent staining for each time point. The cell count (cells/g) and the viability (%) of each class are displayed as box plot graphs, where 5th and 95th percentiles are highlighted with black circles, the medians with solid lines and the means with dotted lines. The data were analyzed using one-way ANOVA on ranks (n = 4 for each time point). (t0: fresh lipoaspirate, t1: 1-month storage, t2: 2-month storage, t3: 3-month storage, t4: 14-month storage, t5: 36-month storage).

**Figure 2 ijms-24-08190-f002:**
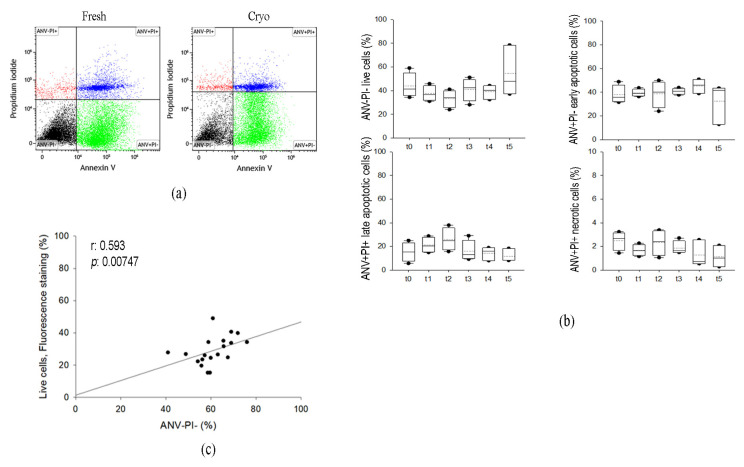
Evaluation of cell viability in the SVF (stromal vascular fraction) of fresh (t0) and cryopreserved adipose tissue (t1–t5) using flow cytometry. (**a**) Representative dot plot of fresh and cryopreserved samples stained with annexin V-FITC and propidium iodide (red dots, ANV-PI+: necrotic cells; blue dots, ANV+PI+: late apoptotic cells; green dots, ANV+PI-: early apoptotic cells; black dots, ANV-PI-: live cells). (**b**) Quantification of cell viability (%) estimated using flow cytometry. The y-axes reported the different populations. The percentages of each class are displayed as box plot graphs where 5th and 95th percentiles are highlighted with black circles, the medians with solid lines and the means with dotted lines. The data were analyzed using one-way ANOVA on ranks and when statistically significant Dunn’s post hoc test was applied (n = 4 for each time point, t0: fresh lipoaspirate, t1: 1-month storage, t2: 2-month storage, t3: 3-month storage, t4: 14-month storage, t5: 36-month storage (**c**) Correlation between the percentage of viable cells estimated using microscopy with fluorescent nuclear staining (y-axis) and flow cytometry (x-axis).

**Figure 3 ijms-24-08190-f003:**
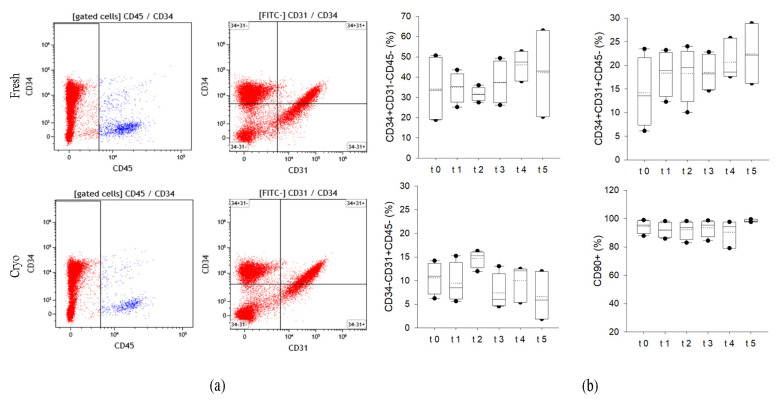
Immunophenotyping of the SVF cells in fresh (t0) and cryopreserved AT (t1−t5) using flow cytometry. (**a**) Representative flow cytometric dot plots of surface markers (CD45, CD34 and CD31) of fresh and cryopreserved lipoaspirates. CD34 vs. CD31 defines the percentage of adipose stromal/stem cells (ASCs) (CD34+CD31-CD45-), endothelial progenitor cells (CD34+CD31+CD45-) and endothelial mature cells (CD34-CD31+CD45-) within SVF. (**b**) Quantification of ASCs, endothelial progenitor and endothelial mature cells contained in SVF. The percentages of each class are displayed as box plot graphs where 5th and 95th percentiles are highlighted with black circles, the medians with solid lines and the means with dotted lines. CD90+ (%) denotes the quantification of CD45-CD31-CD34+ expressing the mesenchymal marker (CD45-CD31-CD34+CD90+). The data were analyzed using one-way ANOVA on ranks. T0: fresh lipoaspirate, t1: 1 month storage, t2: 2-month storage, t3: 3-month storage, t4: 14-month storage, t5: 36-month storage.

**Figure 4 ijms-24-08190-f004:**
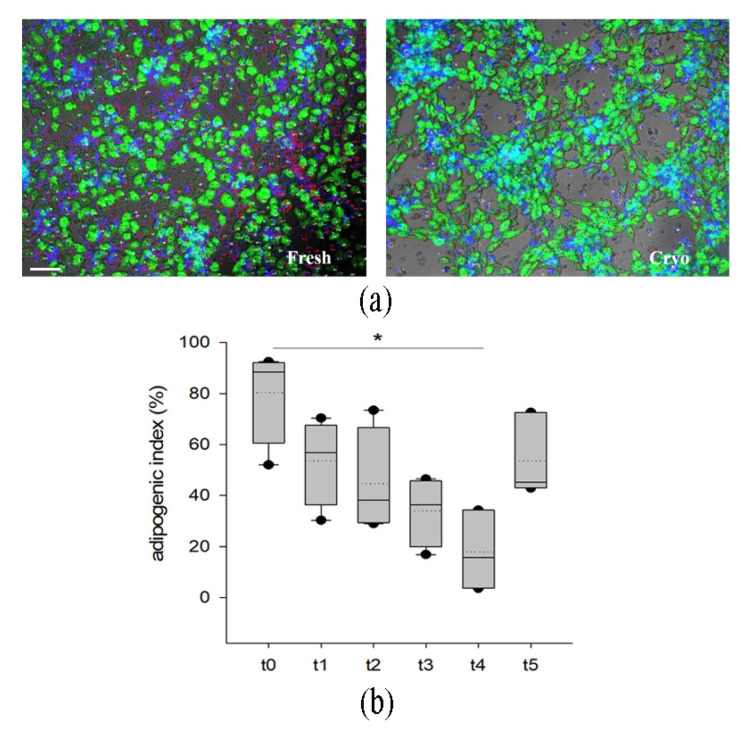
Evaluation of adipogenic differentiation of the SVF (stromal vascular fraction) of fresh (t0) and cryopreserved adipose tissue (t1–t5). Lipid droplets are stained in green using Bodipy 493/503. In (**a**), representative pictures of an in vitro adipogenesis of SVF obtained from fresh and cryopreserved samples. Adipocyte lipid droplets are stained in green; the blue fluorescence highlights nuclei. Magnification 10×, scale bar 100 µm. In (**b**), quantification of the adipogenic index of the SVF. The adipogenic index of each class is displayed as box plot where 5th and 95th percentiles are highlighted with black circles, the medians with solid lines and the means with dotted lines. The data were analyzed using one-way ANOVA on ranks and when statistically significant Dunn’s post hoc test was applied (n = 4 for each time point). * *p* < 0.05. t0: fresh lipoaspirate, t1: 1-month storage, t2: 2-month storage, t3: 3-month storage, t4: 14-month storage, t5: 36-month storage.

**Figure 5 ijms-24-08190-f005:**
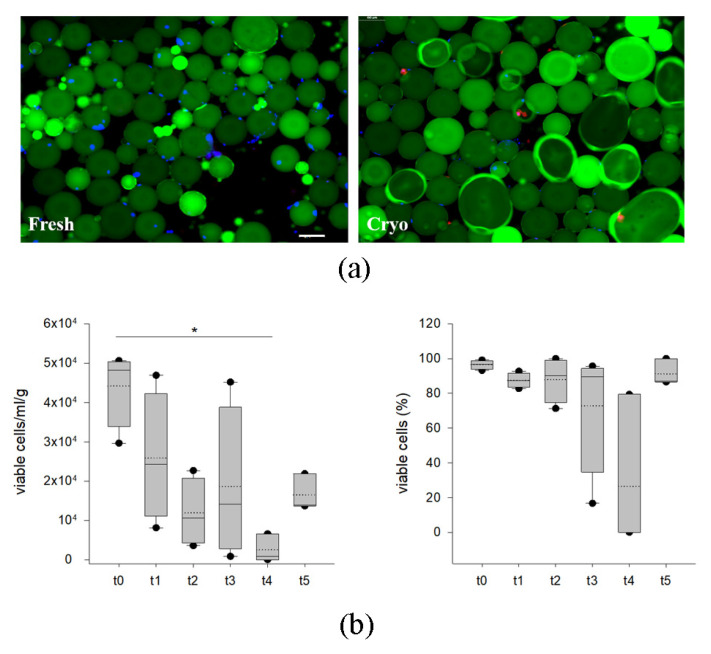
Evaluation of cell viability in the adipocyte fraction of fresh (t0) and cryopreserved AT (t1–t5) using microscopy. In (**a**), representative pictures of a hemocytometer loaded with cell suspension ReadyProbes Cell Viability solution, obtained from fresh and cryopreserved samples. Lipid droplets are stained in green using Bodipy 493/503. Hoechst 33342 (blue) stains all nuclei and propidium iodide stains nuclei of cells with compromised plasma membrane integrity. Magnification 10X, scale bar 100µm. (**b**) Number of viable adipocytes normalized by sample weight (g), calculated for each time point using nuclear fluorescent staining. Percentage of viable adipocytes for each time point calculated using nuclear fluorescent staining. The number of adipocytes and their viability (%) are displayed as box plot graphs where 5th and 95th percentiles are highlighted using black circles, the medians using solid lines and the means using dotted lines. The data were analyzed using one-way ANOVA on ranks and when statistically significant Dunn’s post hoc test was applied (n = 4 for each time point). * *p* < 0.05. t0: fresh lipoaspirate, t1: 1-month storage, t2: 2-month storage, t3: 3-month storage, t4: 14-month storage, t5: 36-month storage.

**Figure 6 ijms-24-08190-f006:**
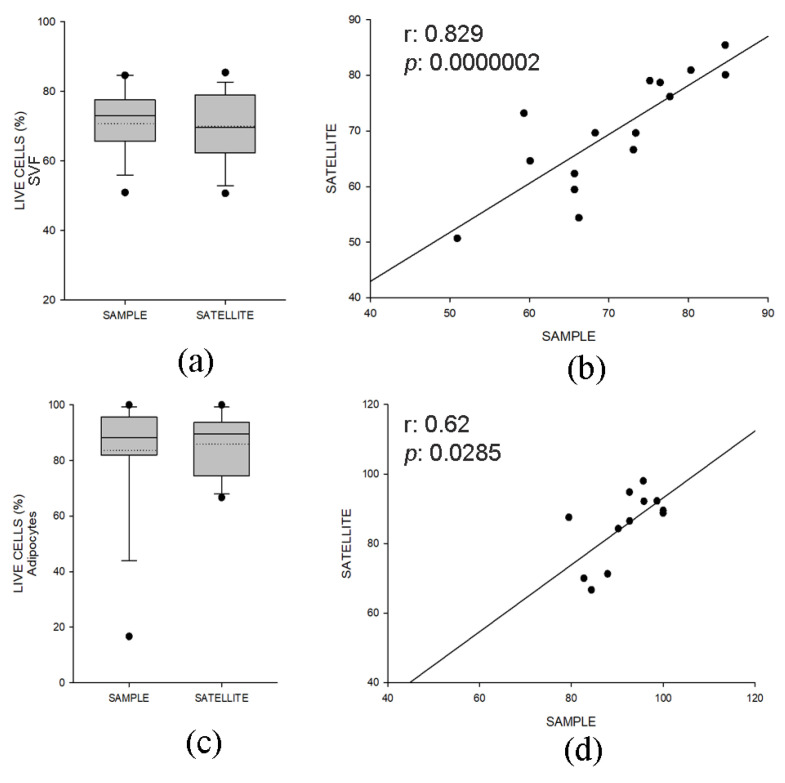
Evaluation of cell viability in cryopreserved AT samples and their paired quality controls (satellite) (n° of samples = 15). In (**a**), percentage of live cells (stromal vascular fraction, SVF) estimated by immunofluorescence in fat samples and their paired quality controls (satellite). Data (n = 15) are displayed as box plot graphs where 5th and 95th percentiles are highlighted using black circles, the medians using solid lines and the means using dotted lines. In (**b**), correlation between the percent of viable cells (SVF) quantified in adipose samples (x-axis) and paired quality controls (satellite). In (**c**), percentage of live cells (adipocytes, AD) estimated by immunofluorescence in fat samples and their paired quality controls (satellite). Data (n = 12) are displayed as box plot graphs where 5th and 95th percentiles are highlighted using black circles, the medians using solid lines and the means using dotted lines. In (**d**), correlation between the percent of viable cells (AD) quantified in adipose samples (x-axis) and paired quality controls (satellite). Spearman’s correlation coefficient (r) and significance (*P*) were reported in the plots.

**Figure 7 ijms-24-08190-f007:**
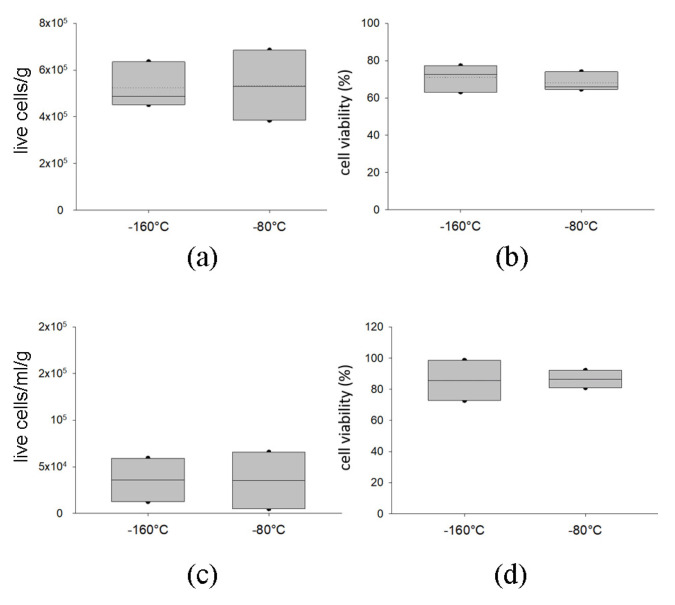
Evaluation of the effects of the delivery temperature on cell number and cell viability of SVF and AD in cryopreserved samples. (**a**) Number of viable cells normalized by sample weight (g), and in (**b**), percentage of viable SVF calculated using nuclear fluorescent staining on samples thawed readily after the cryopreservation in vapor phase liquid nitrogen (−160 °C) or following 24 h storage with dry ice (−80 °C) after cryopreservation. In (**c**), number of viable cells normalized by sample weight (g) and (**d**) percentage of viable AD calculated using nuclear fluorescent staining on samples thawed readily after the cryopreservation in vapor phase liquid nitrogen (−160 °C) or following 24 h storage with dry ice (−80 °C) after cryopreservation. The cell count (cells/g or cells/mL/g) and the viability (%) of each class are displayed as box plot graphs, where 5th and 95th percentiles are highlighted using black circles, the medians using solid lines and the means using dotted lines. The data were analyzed using one-way ANOVA on ranks (n = 3 for each condition).

**Figure 8 ijms-24-08190-f008:**
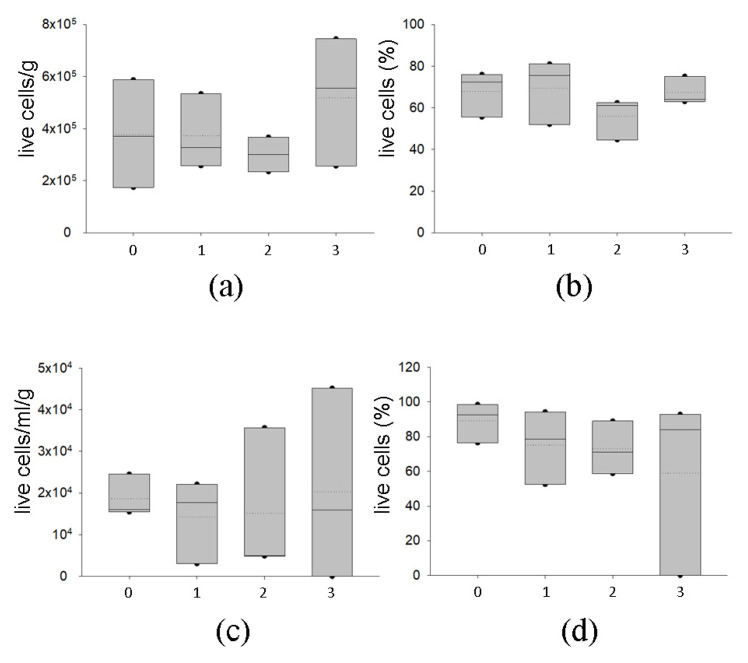
Evaluation of the effects of short-term storage at −80 °C on cryopreserved samples. In (**a**), number of viable SVF cells normalized by sample weight (g) and (**b**) percentage of viable SVF. (**c**) Number of viable AD cells normalized by sample weight (g) and (**d**) percentage of viable AD. AT samples were kept 1 month in vapor phase liquid nitrogen and then thawed (0) or further stored at −80 °C for 1, 2 or 3 months before subsequent analyses. Cell count and viability were estimated using nuclear fluorescent stains; each class of data are displayed as box plot graphs, where 5th and 95th percentiles are highlighted using black circles, the medians using solid lines and the means using dotted lines. The data were analyzed using one-way ANOVA on ranks (n = 3 for each time point).

**Figure 9 ijms-24-08190-f009:**
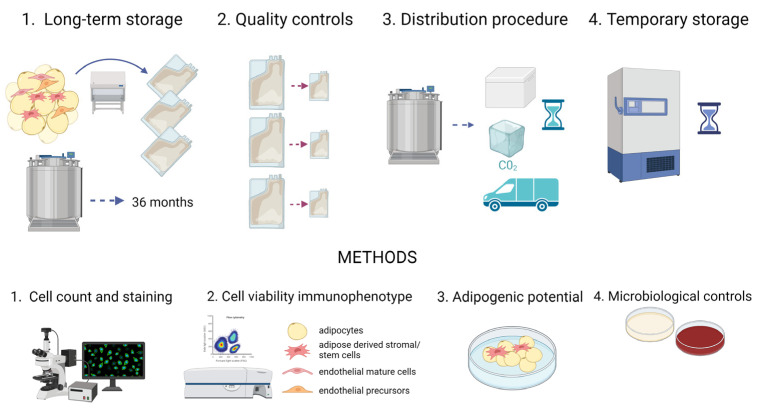
Representative schematic procedure of the study. In the upper part, lipoaspirates were collected from donors, and samples were stored in vapor phase liquid nitrogen to test the long-term storage (1), establish a quality control for cryopreserved sample (2), evaluate the distribution procedure in dry ice (3) and the temporary storage at −80 °C (4). Below, a brief description of the methodologies used for the analyses. Created with BioRender.com.

**Table 1 ijms-24-08190-t001:** Antibody panel for the immunophenotyping of SVF cells using flow cytometry.

	FITC	PE	PerCP.Cy5.5
Unstained	-	-	-
Negative	IgG1	IgG1	IgG1
Sample 1	CD45	CD31	CD34
Sample 2	CD45 CD31	CD90	CD34

## Data Availability

Data are contained within the paper and its Appendix A.

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
