# Peer review of "Characterization of Human Subcutaneous Adipose Tissue and Validation of the Banking Procedure for Autologous Transplantation"

_ijms, 2023, doi:10.3390/ijms24098190_

Round 1

Reviewer 1 Report

The Authors of the manuscript entitled "Characterization of human subcutaneous adipose tissue and validation of banking procedure for autologous transplantation" aimed to widely characterize the cryopreserved adipose tissue and to validate its storage up to three years for clinical application. The subject of the study seems to be interesting for practical reasons. However, there are some points of concern in terms of the manuscript quality that need to be carefully considered.

The major weaknesses with the submitted manuscript are, as follows:

1.      Contrary to the title, the characterization of human adipose tissue is not exhaustive. It is based on only few markers, or even one (CD34), not fully representative for adipose tissue stem cells.

2.      The expression of the hematopoietic marker CD34 by ADSC has been questioned. It was different between donors and their passages that accounted from about 1 to more than 23 percent. Several studies found that freshly isolated ADSC expressed CD34, but all of these studies and several other on mesenchymal stem cells observed disappearance of CD34 expression when cells were propagated in culture.

3.      The results of Microbiological tests are indicated in the manuscript but the tests and its results are not described.

4.      Only one method was used for the cell viability assessment, and it was not confirmed by another method.

5.      Authors write about ‘different molecular methods’, as methods they have used in this study, but unfortunately such methods (molecular) were not actually applied.

6.      The conclusion is too much extended. This study is not a clinical research and an information on two patients can not be a part of it.

Minor remarks

1.      The abbreviation  ASC (it can be also Adult Stem Cells) should be substituted by ADSC (Adipose Tissue Stem Cells), what is more appropriate name for AT derived stem cells.

2.      Figure 4: there is no images showing positive control, negative control, and scale bars are omitted in pictures ‘a’ and ‘b’.

3.      It is not clear how many samples were used in different assays and time points. Moreover, AT was collected from seven donors, but ‘n’ in figure captions is given as 4. What was a positive control for Annexin V staining?

4.      It is not explained why the Authors changed an order of testing procedures during cell isolation, as compared to the original protocol (page 14, modified [54]).

5.      Antobodies were poorly characterized (Table 1). There is no information about the catalog numer, dilution, host etc.

6.      In Discussion section there are direct references to the Results and some of illustrations (page 11), it should not happen in this section.

Reviewer 2 Report

-The authors are recommended to discuss the dark side as well as the bright side of the AT heterogeneity.

- In Materials and Methods section: The authors need to provide the reference that related to Rodbell’s modification protocol for cell isolation.

-The authors need to give a detailed explanation of how they calculate cell yield per gram of tissue. 

-What type of microbiological tests applied in this study? The authors need to explain about the test and their results.

-The authors performed the decontamination process for at least 12..is it enough for the antibiotics course and any reference? And why the authors specifically chose those antibiotics?

Is there any information about the donors' health condition, gender, age etc?
